# Effects of Leaf Removal on Free and Glycoconjugate Aromas of Skins and Pulps of Two Italian Red Grapevine Varieties

**DOI:** 10.3390/foods12193661

**Published:** 2023-10-04

**Authors:** Paola Piombino, Elisabetta Pittari, Alessandro Genovese, Andrea Bellincontro, Osvaldo Failla, Luigi Moio

**Affiliations:** 1Department of Agricultural Sciences, Division of Vine and Wine Sciences, University of Naples Federico II, 83100 Avellino, Italy; elisabetta.pittari@unina.it (E.P.); moio@unina.it (L.M.); 2Division of Food Science and Technology, Department of Agricultural Sciences, University of Naples Federico II, 80055 Portici, Italy; alessandro.genovese@unina.it; 3Dipartimento per la Innovazione nei Sistemi Biologici, Agroalimentari e Forestali (DIBAF), University of Tuscia, 01100 Viterbo, Italy; bellin@unitus.it; 4Department of Agricultural and Environmental Sciences, University of Milan, 20133 Milano, Italy; osvaldo.failla@unimi.it

**Keywords:** leaf thinning, fruit set, berries touch, Italian red grapevine varieties, skin and juice, free and bound VOCs, SPE-GC/MS

## Abstract

Leaf removal is a cultural practice mainly aimed at improving cluster zone microclimates and impacting primary and secondary metabolites, such as volatiles. This research aimed to assess the impact of defoliation on free and glycosylated aromas of a neutral (‘Nebbiolo’) and a semi-aromatic (‘Aleatico’) red variety. Defoliation was performed at fruit set (BBCH 71) and, for ‘Nebbiolo’, also at berries touch (BBCH 81) phenological stages. Skins and pulps were separately analyzed by Solid Phase Extraction–Gas Chromatography/Mass Spectrometry. Results showed that the response to defoliation was variety-dependent. For ‘Nebbiolo’, especially when performed at the berries’ touch stage, defoliation had a significant effect on the accumulation of free volatiles and glycosidic precursors. Differently, free and bound ‘Aleatico’ volatiles were less impacted by defoliation. Interestingly, in both grapevine varieties, defoliation significantly enhanced the accumulation of aroma precursors in grapes’ skins, which is of particular relevance for red wine production and their aging potential. Moreover, results could be helpful for the management of grape quality, as defoliation is currently considered as a strategy to address climate change issues.

## 1. Introduction

In the production of high-quality wines, canopy/crop-management practices (i.e., shoot trimming, pruning, cluster thinning, irrigation, and defoliation) are widely applied by grape growers and winemakers to improve berry composition. 

Among these techniques, cluster zone leaf removal, also known as leaf thinning or defoliation, is a cultural practice mainly aimed at improving canopy and cluster zone microclimates and berry quality [1,2]. It could be applied before flowering through veraison, and it consists of removing leaves from around the fruiting zone.

In the past few years, researchers have investigated the effects of timing and severity of defoliation on grape and wine characteristics. Mainly depending on variety, environment, and defoliation timing, the management of leaf removal affects several grape morphological and physical–chemical characteristics that are important from a technological and a sensory point of view. Indeed, with leaf removal, viticulturists aim to optimize sunlight exposure, air circulation, and spray penetration, having an influence on grape development and berries’ primary and secondary metabolites, which, in turn, may improve the aroma and flavor development of the berries [3]. 

Defoliation directly affects grapes’ sunlight exposure: increased sunlight exposure on grape clusters can affect the synthesis of sensory active compounds such as polyphenols (responsible for red wine color and astringency) [4] and volatiles, both in free and glycoconjugate forms [3] In fact, as early as the late 1990s, researchers showed the beneficial effects of defoliation on the volatile composition of grapes. Monoterpenes and/or glycoconjugates volatiles have been observed to be responsive to increases in fruit exposure and to leaf removal and cluster thinning [5,6,7,8]. Reynolds and Wardle [5] studied Gewurztraminer vines subjected over three seasons to seven canopy manipulation practices, among which included basal leaf removal. The authors concluded that the elimination of apparently superfluous sinks, such as shoot tips, lateral shoots, and basal leaves, moderately reduced the canopy density and increased the concentration of both free and potential volatile terpenes in the fruit. Moreover, Hunter et al. [9] showed that regardless of the severity of defoliation or developmental stage, when leaf removal was applied, the wine quality of Cabernet Sauvignon was significantly improved by an increase in varietal expression and overall wine quality. Overall, across the past 10 years, the studies [10,11,12,13,14,15,16,17,18,19,20] showed that leaf removal affected the volatile profile of berries obtained from different cultivars, leading to an increase in free and glycosylated aroma compounds, such as monoterpenes (i.e., beta-linalool, alpha-terpineol, and nerol) and C13-norisoprenoids (i.e., beta-damascenone).

Moreover, defoliation can also decrease disease pressure, particularly in compacted cluster varieties: better air circulation can help reduce the risks of fungal diseases due to a reduction in the number of set berries and, in turn, to looser clusters that are less susceptible to Botrytis rot [21], which is also interesting for grapes undergoing dehydration for sweet wine production. 

In this context, the main aim of the current research was to assess the impact of defoliation practices on grape-free and glycosylated volatile compounds. For the first time, the investigations were conducted by analyzing separately the different portions of the berry: the skin and the pulp. Moreover, we considered both a neutral and a semi-aromatic variety, namely the two Italian ‘Nebbiolo’ and ‘Aleatico’ red grapes, respectively. From ‘Nebbiolo’ grapes, the high-quality and world-renowned dry wines characterized by specific in-mouth features [22], such as Barolo and Barbaresco DOCG (Piemonte) and Sforzato di Valtellina DOCG (Lombardia), one of the few passito red wines, are produced. Another dessert red wine (Aleatico Passito dell’Elba DOCG) is traditionally produced from the ‘Aleatico’ grapevine variety, which is particularly cultivated on Elba Island (Tuscany).

## 2. Materials and Methods

### 2.1. Defoliation and Sampling Protocol

*Vitis vinifera* L. cv. ‘Nebbiolo’ and ‘Aleatico’ grapevines were grown in Valtellina area and Lazio region, respectively. Procedures for sampling and defoliation were carried out as recently detailed [23]. Briefly, according to the extended BBCH scale [24], defoliation was performed at two phenological stages for ‘Nebbiolo’ grapes: early defoliation at fruit set (BBCH 71: D1), and late defoliation at the beginning of bunch closure, berries touch (BBCH 81: D2); while only one treatment was applied to ‘Aleatico’ grapes, at fruit set (BBCH 71: D1). D1 was applied in mid-June and D2 in mid-July. In detail, basal leaves shading the bunches were manually removed to increase bunch exposure to sunlight without excessive limitation of photosynthate availability. To achieve this goal, at least 1 m^2^ of leaf area per kg of grape production was guaranteed in all the treatments. ‘Nebbiolo’ vine blocks were distributed in the upper, middle, and lower parts of the vineyard. For the ND, D1, and D2 treatments, 583, 583, and 579 vines, respectively, were selected in a completely randomized block design (comprising 9, 8, and 8 blocks for ND, D1, and D2, respectively). In each of the upper, middle, and lower parts of the vineyards, 8 rows (one row per block, 80 vines per treatment) were designated for measurement of crop yield (the number of bunches) per vine and the average mass of bunches and berries (data not shown). For ‘Aleatico’ grapevines, considering the E-W vineyard exposition able to ensure the same light intervals on the rows, an entire vineyard block consisting of 10 rows was designed for the experiment. Two external rows (one for side) were excluded from the test, while the two rows just adjacent to one and the opposite side were designed for ND. Finally, the four central rows of the vineyard were destined for D1 treatment. Fifty vines selected in the middle part of each row were designated for leaf removal (D1) treatment or as control (ND), as well as for making the same carpometric determinations reported before (data not shown). A total number of 200 vines for both ND and D1 treatment were considered. Canopy assessment to determine the proportion of gaps, leaf layer number, and proportion of interior leaves and fruit was achieved via the ‘point quadrant’ method [25] adapted to grapevines. The sugar concentration of berries was similar among treatments; thus, the harvest of all treatments occurred within 2 days in mid-October (BBCH 89), when berries reached a TSS of 23 ± 1° Brix, in light of the production of sweet wines from off-vine dehydrated grapes.

### 2.2. Volatile Organic Compounds (VOCs) Analysis

Volatile Organic Compounds (VOCs) were analyzed as recently reported [23]. Briefly, 400 g of berries for each treatment were processed and analyzed with a method previously optimized by Genovese et al. [26] and used for both grapes [27] and wine analyses [28]. The berries were prepared by removing their peduncles, and then the skins and pulp were meticulously separated using tweezers. To prevent oxidation, the skins were immediately placed in bottles containing a must-like buffer solution (5 g/L of tartaric acid, 10 g/L of PVPP and 2 g/L of sodium azide and was pH adjusted to 3.2 with 1 N NaOH) for extraction. The skins were stirred for 24 h at 20 °C in the absence of light and then centrifuged at 10,000× *g* for 20 min at 20 °C. After separating the skins, the deseeded pulp was homogenized using a CJ60 homogenizer (Black & Decker, Towson, MD, USA) for 2 min, with the addition of 2 g/L of sodium azide. Subsequently, it was centrifuged at 10,000× *g* for 10 min at 10 °C using an ALC 4239R centrifuge (Daihan Scientific, Wonju, South Korea). The resulting liquids were filtered through cellulose paper to obtain sample solutions, which were then stored at −20 °C until analysis (two replicates). For each grape sample, 50 mL of the sample solution was spiked with 250 μL of 2-octanol (200 mg/L in methanol) and passed through a C18 reversed-phase solid-phase extraction (SPE) column (1-g C18 cartridge, Phenomenex, Torrance, CA, USA). The column was rinsed with Milli-Q water, and the adsorbed volatiles were eluted with dichloromethane, while bound volatiles were eluted with methanol. The methanol was evaporated under reduced pressure at 37 °C, and the residue was dissolved in 5 mL of citrate–phosphate buffer (pH 5.0) containing 80 mg of Rapidase AR 2000 pectolytic enzyme with secondary glycosidase activities (DSM, Delft, Holland) before being incubated for 16 h at 40 °C. The volatiles released by enzymatic hydrolysis were eluted with dichloromethane on preconditioned C18 cartridges after the addition of 250 μL of 2-octanol as an internal standard. The extracts were dried over Na_2_SO_4_ and finally concentrated to 50 μL under a N_2_ stream. Both free and bound VOCs were analyzed by GC/MS and GC/flame ionization detector (GC/FID), following a method previously described [29]. For each sample, the SPE procedure was duplicated. GC/MS analysis was performed using a Shimadzu GC/MS-QP2010 mass spectrometer (Shimadzu, Kyoto, Japan) equipped with a split/splitless injector and a DB-WAX column (60 m × 0.250 i.d., 0.25 μm film thickness) (J&W Scientific, Folsom, CA, USA). The temperature program involved an initial 40 °C for 5 min, followed by an increase to 220 °C at 2 °C/min, and then held at 220 °C for 20 min. Helium was used as the carrier gas at a flow rate of 1.02 mL/min. The samples (approximately 1.2 μL) were injected in splitless mode, with the injector port and ion source maintained at 250 °C and 230 °C, respectively. Positive electron impact spectra were recorded in the range of *m*/*z* 33–350. Compound identification was confirmed by injecting pure standards and comparing retention times and MS spectra with those in the NIST 2.0 library. For GC/FID analysis, an Agilent 7890 A chromatograph equipped with a split/splitless injector and a J&W DB Wax column was employed. The same temperature program used for GC/MS analysis was followed. Helium served as the carrier gas with a flow rate of 2.20 mL/min. Two replicates of each aroma extract (1.2 μL) were injected in splitless mode, and the detector and injector were maintained at 250 °C. Volatile compounds were quantified using calibration curves, with peak areas normalized relative to the internal standard peak area and interpolated using the calibration curve. Calibration graphs were generated by analyzing a blank solution as previously described [26] and spiking it with known amounts of each analyte and internal standard. The solution was then diluted to obtain calibration points for each analyte, with the concentration range aligning with values typically found in Italian grape cultivars [27]. The linear regression coefficient (r^2^) for each volatile compound was ≥0.9918, consistent with previous literature. Dry mass (DM) was used instead of fresh mass to avoid concentration effects during computation, considering water loss through a moisture content measurement conducted via an oven-dry method.

### 2.3. Data Processing

VOCs chemical data (mean values obtained by 2 extractions * 2 injections; n = 4) were treated by an analysis of variance ANOVA (Tukey; *p* < 0.05) to test significant differences among the treatments. Computations were made using XLStat 2012.6.02 (Addinsoft Corp., Paris, France).

## 3. Results and Discussion

In Figure 1, the concentration (expressed in µg/kg of berries) of free and glycoconjugate volatiles and the total amount for each class of identified VOCs of ‘Nebbiolo’ skin and pulp is represented. Six classes of free volatiles and four of glycoconjugate aromas were identified in ‘Nebbiolo’ grapes. Specifically, GC/MS analyses allowed the detection of C6 compounds, alcohols, terpenes, and benzenoids in both free and bound forms, while few esters (butyl acetate) and acids (hexanoic acid) were present only in free form. 

Except for total free terpenes and total glycoconjugate alcohols in the pulp, all the other identified chemical classes showed significant differences in both the skin (free VOCs generally decreased in D1 and then increased in D2; bound VOCs generally increased in D1 and then decreased in D2) and the pulp (generally increased constantly from the control to D2 in both free and bound forms), suggesting an impact of defoliation and its application timing on the volatile profile of this grapevine variety. 

Starting from the volatile composition of the skin (Figure 1a,b), both in free and bound forms, defoliation at fruit set (D1) had the greatest impact and resulted in opposite behaviors. In particular, D1 samples showed the lowest free and highest bound VOC concentrations, both in terms of total amount and for all the identified classes of volatiles. This suggests that the response to leaf removal applied in the early phenological phase of the fruit set is more evident than that conducted in the later berries touch phase, which in the case of ‘Nebbiolo’ also corresponds to veraison. 

Moreover, considering the free form of skin aromas, except for the ester butyl acetate, defoliation at berries touch (D2) led to higher contents of all the other identified classes of volatiles, significant for C6 compounds, terpenes, and benzenoids. Finally, regarding skin glycoconjugate aromas, except for alcohols, for all the other groups of VOCs, the same behavior can be observed: the sample defoliated at berries touch (D2) was richer in bound VOCs compared to the control sample (ND), but lower than the early defoliated (D1).

Moving to ‘Nebbiolo’ volatile composition of the pulp (Figure 1c,d), a common trend for both free and glycoconjugate forms can be observed: except for the free ester butyl acetate and the free hexanoic acid, the non-defoliated samples (ND) showed the lowest concentrations both in terms of total amount and for all the identified classes of VOCs. Furthermore, regarding the glycoconjugate aromas, differently from the skin, the sample defoliated at berries touch (D2) showed the highest concentrations of nearly all the identified classes of VOCs (C6 compounds, terpenes, and benzenoids) and consequently of the total bound volatile composition. 

In the three samples, considering the chemical classes computed on the whole grape berries (skin + pulp), except for C6 compounds (more abundant in free form than in glycoconjugate form), butyl acetate and hexanoic acid (detected only in free form), the total concentration of bound form of the other chemical classes (i.e., alcohols, terpenes, and benzenoids) was between 2 and 16 times that of free aroma. The higher amount of C6 compounds in free form could be linked to their role. Indeed, the free form of green leaf volatiles serves as interplant signals, allowing communication between plants [30], and they are also defense compounds when plants are attacked by herbivores [31].

However, considering the total amount of the identified VOCs (all chemical classes of the whole grape berries: skin + pulp), the non-defoliated samples showed 1.20 times higher concentrations of VOCs in free form (9706 µg/kg) than those in bound form (7940 µg/kg). In contrast, samples where defoliation treatments were applied (D1 and D2) showed 1.32–2.50 times higher concentrations of volatiles in bound form (D1: 18,685 µg/kg; D2: 15,900 µg/kg) than those in free form (D1: 7422 µg/kg; D2: 11,991 µg/kg). Therefore, the defoliation of ‘Nebbiolo’ grapes led to an increase in glycosylation, which is of particular oenological interest. Indeed, this accumulation is particularly evident on the skin, and this is an interesting result considering the importance of skin composition in red wine vinification. Furthermore, the increase in glycosylation corresponds to a greater aromatic potential that can be expressed during fermentation and wine aging by the hydrolysis of these precursors and the release of odorous aglycones. Therefore, this could be of interest in the production of long-aging wines usually obtained by ‘Nebbiolo’ grapes.

In Table 1 and Table 2, the detailed free and glycoconjugate volatile composition of the different portions of ‘Nebbiolo’ grape berries belonging to the specific chemical classes is given. Quantitative GC/MS analyses allowed the identification and quantification of 23 free VOCs (Table 1) and 20 glycoconjugate VOCs (Table 2). Significant differences were found for nearly all the detected free and bound volatile compounds; only free alpha-terpineol in the pulp (Table 1b) and the two glycoconjugate 3-methyl-1-butanol in the skin (Table 2a) and 1-heptanol in the pulp (Table 2b) did not show significant differences between the three defoliation treatments. 

Similar behaviors were previously observed in response to leaf removal treatments studying Aglianico [19], Malvasia [16], Merlot [12,18], Pinot noir [10,11], Sauvignon blanc [13,17], Semillon [15], Tempranillo [14], and Xynisteri [20] cultivars, where authors hypothesized that these increased amounts following defoliation could have been due to the increased light exposure, since VOCs emissions are known to increase in response to both biotic and abiotic stresses, including temperatures and light exposure [32]. For example, it has been observed that in those stress conditions, plants alter membrane fluidity, releasing significantly higher amounts of alpha-linolenic acid from membrane lipids, and therefore increasing lipoxygenase (LOX) activity, which in turn leads to higher synthesis of C6 compounds [33,34]. Of particular oenological interest is the increased terpenes content observed in exposed grapevines due to defoliation treatments since these are important varietal aroma compounds with olfactory properties highly appreciated by consumers. This behavior has already been observed in the literature [13,35]. However, completing the studies previously conducted on the whole grape berries, our data showed that this accumulation mainly occurred in the skins, supporting the hypothesis reported on “Malbec” grape berries that these compounds could play a role in protecting plants from environmental stress conditions (i.e., heightened UV-B radiation exposure) [35]. In fact, as also observed by Loreto and Schnitzler [36], monoterpenes could be able to stabilize chloroplastic membranes, thereby fending off oxidative damage. Even if the underlying mechanism is still not clear, volatile terpenes are established to possess antioxidant properties, potentially impacting membrane functions and stability due to their lipophilic nature. Monoterpenes could cooperate with carotenoids to heighten photoprotection under abiotic stress conditions, such as heightened light and temperature, contributing to the regulation of oxidative stress equilibrium [37].

Figure 2 shows the behavior of free and glycoconjugate aromas of skin and pulp of ‘Aleatico’ grapes in terms of the total amount and global amount for each chemical class of identified VOCs. The detailed free and glycoconjugate volatile composition of the different portions of ‘Aleatico’ grape berries is represented in Table 3 and Table 4, respectively. 

As expected for a semi-aromatic grape variety, a greater number of free and bound volatile molecules has been identified in ‘Aleatico’ grapes compared to ‘Nebbiolo’, which is a neutral grape. GC-/MS analyses allowed the detection of 35 free VOCs (Table 3) and 46 glycoconjugate VOCs (Table 4) belonging to seven chemical classes: C6 compounds, alcohols, terpenes, benzenoids, esters, acids, and other compounds (the free furfural and the glycoconjugate 3-hydroxy-beta-damascone). In the specific case of terpene compounds, 9 free and 16 bound terpenes were identified and quantified in ‘Aleatico’ grapes, in comparison with the 2 free and 3 bound detected in ‘Nebbiolo’ grapes. 

Very few significant differences were found between the control and the defoliated sample in terms of total amount of VOCs and global amount for each chemical class (Figure 2), suggesting that defoliation at fruit set had a scarce impact on ‘Aleatico’ grapes’ volatile composition. However, regarding skin VOCs (Figure 2a,b), both in terms of the total free and total bound, such as in the case of ‘Nebbiolo’ grapes, a higher concentration in the defoliated sample was observed. The differences were significant only for the glycoconjugate form, mainly due to a significantly higher amount of total C6 compounds and total benzenoids. Even if the total glycoconjugate terpenes in the skins showed no significant differences between the two samples, in line with data reported in the literature [10,11,13,20], some of these important aromas were significantly higher in D1 compared to the ND sample (Table 4a): trans-linalool oxide, cis- and trans-citral, citronellol, cis- and trans-geraniol. As already stated, this increased amount could be due to the increased light exposure and temperature conditions of the bunches in the defoliated grapevines [32]. Regarding pulp VOCs (Figure 2c,d), an opposite trend can be observed: total free and total bound showed lower concentrations in the D1 sample, significant only for the glycoconjugate form. Considering the specific classes of chemical compounds in the pulp, free terpenes, benzenoids, and furfural (Figure 2c) and bound C6 compounds, alcohols, terpenes, benzenoids, and the hexanoic acid (Figure 2d) showed significant differences. The observed trends (free and bound VOCs increase in the skin and decrease in the pulp) could also be related to previous results on ‘Aleatico’ grapes, showing that defoliation around the fruit set can induce a positive effect by reducing berry dimension as well as enhancing the skin thickness [38], favorable characteristics for grapes destinated to dehydration for sweet wine vinification [39]. 

In both treatments (control and defoliation at fruit set), considering all chemical classes of the whole grape berries (skin + pulp), the total concentration of the bound form (ND: 78,902 µg/kg; D1: 88,048 µg/kg) was about 2.5 times that of the free one (ND: 33,266 µg/kg; D1: 33,642 µg/kg). This is mainly due to terpenes and benzenoids, which in the bound form in the whole grape berries (skin + pulp) were between 6 and 10 times that of the free form. In contrast, the other chemical classes identified (i.e., C6 compounds, esters, and acids) showed higher concentrations as free form. 

These results confirm that the VOC pattern of ‘Aleatico’ grapes is mainly characterized by terpenes and benzenoids, confirming the semi-aromatic nature of these grapes. 

The results suggest that the defoliation effects on VOCs and their precursors are variety-dependent as ‘Aleatico’ and ‘Nebbiolo’ show different trends under the same leaf removal treatment at fruit set. Part of this diversity could be ascribable to the different cultivation areas, but genetic reasons could also play a role. The two grape varieties are characterized by similar phenological features [40], but ‘Nebbiolo’ is a neutral grape while ‘Aleatico’ is a semi-aromatic one, as testified by a total free monoterpene concentration that can reach 6 mg/L [41]. Enzyme differences were also reported; for example, alcohol dehydrogenase activity (ADH) was found to be 100 µmol NADH/g dry weight in ‘Aleatico’ grapes [42], while 85 in ‘Nebbiolo’ [23]. This difference could impact the volatiles’ pattern of the investigated grapevines and possibly their response to defoliation, which was previously documented as significant in ‘Nebbiolo’ [23]. The ADH indeed plays a role in the biosynthesis of an important group of aroma volatiles, C6-derivative compounds, including different aldehydes, alcohols, and esters [43,44,45]. Specific experiments are necessary to verify the observed trends and the underlying reasons.

## 4. Conclusions

The impact of defoliation on free and glycoconjugate aromas of the ‘Nebbiolo’ and ‘Aleatico’ grapevine varieties has been studied. Even if data are limited to one season, for the first time, the different portions of the berry, namely the skin and the pulp, have been analyzed separately. This approach provided a first detailed analysis of the impact of this practice on the spectrum of volatiles of a neutral and a semi-aromatic red grape variety. 

Results suggested that the response to defoliation treatments is variety-dependent. In the case of ‘Nebbiolo’ grapes, defoliation and, particularly, that conducted at the berries touch (D2) can have a significant effect on the accumulation of free volatiles and glycosidic precursors, therefore suggesting that this practice could be considered for managing the volatile profile of these grapes. Differently, at fruit set (D1), ‘Aleatico’ grapes showed a less-evident impact of leaf removal on VOC accumulation compared to ‘Nebbiolo’. Globally, the effect of defoliation at fruit set (D1) is more evident in the volatile composition of the neutral grape compared to the semi-aromatic one, likely also because small variations may be more detectable when they occur in a less concentrated matrix.

Despite this difference, of particular interest is that leaf removal favored the glycosylation of skin aromas in both varieties, which, according to the literature, could be an indirect effect of increased temperatures and light exposure of bunches. This represents an interesting result considering the importance of skin composition in red wine vinification and the potential oenological role of the glycosidic precursors in the shelf-life of wine aroma.

Defoliation is currently proposed as a strategy to manage challenging environmental conditions in the vineyard due to climate change. Therefore, these results could be useful in an optic of managing grape quality in this context and also to optimize human and economic resources already in the field. 

## Figures and Tables

**Figure 1 foods-12-03661-f001:**
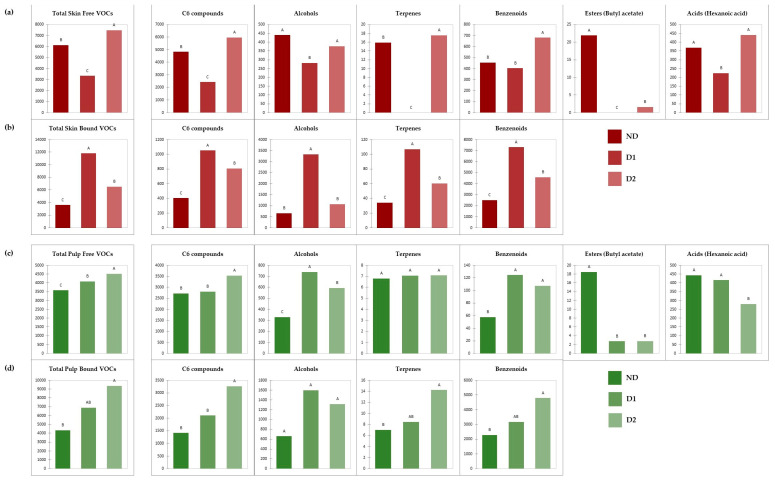
Effect of defoliation on (**a**) Skin free VOCs; (**b**) Skin bound VOCs; (**c**) Pulp free VOCs; (**d**) Pulp bound VOCs of ‘Nebbiolo’ grapes. Different letters refer to significant differences (Tukey; *p* < 0.05). All data are expressed in µg/kg of berries.

**Figure 2 foods-12-03661-f002:**
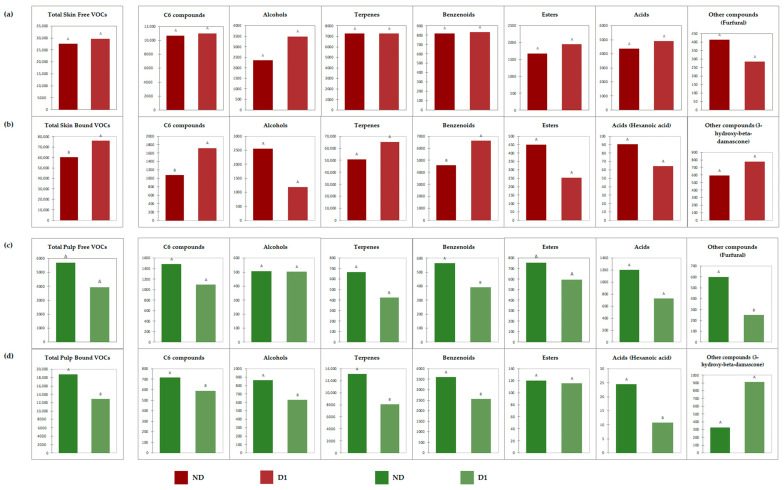
Effect of defoliation on (**a**) Skin free VOCs; (**b**) Skin bound VOCs; (**c**) Pulp free VOCs; (**d**) Pulp bound VOCs of ‘Aleatico’ grapes. Different letters refer to significant differences (Tukey; *p* < 0.05). All data are expressed in µg/kg of berries.

**Table 1 foods-12-03661-t001:** ‘Nebbiolo’ free VOCs detected in (a) skin and (b) pulp of ND (not defoliated/control), D1 (defoliated at fruit set—BBCH 71), and D2 (defoliated at berries touch—BBCH 81) samples. Mean concentration (expressed in µg/kg of berries) and standard deviation for each compound are reported.

	(a) SKIN				(b) PULP			
	ND	D1	D2	Sign.	ND	D1	D2	Sign.
*C6 compounds*								
hexanal	2453.3 ± 151.9 A	1187.3 ± 85.6 B	2745.8 ± 292.0 A	*	1460.9 ± 40.8 B	1364.2 ± 90.7 B	1821.9 ± 42.6 A	*
trans 2-hexenal	1344.7 ± 62.7 B	834.7 ± 84.6 C	2285.5 ± 63.8 A	*	214.7 ± 17.4 C	548.9 ± 63.1 B	846.6 ± 107.5 A	*
1-hexanol	364.2 ± 21.4 A	173.0 ± 9.1 B	351.5 ± 32.7 A	*	351.0 ± 12.6 A	345.7 ± 13.8 AB	312.2 ± 15.5 B	*
cis 3-hexen-1-ol	4.8 ± 0.1 B	4.8 ± 0.1 B	5.1 ± 0.1 A	*	0.2 ± 0.0 C	1.7 ± 0.2 AB	1.3 ± 0.1 B	*
trans 3-hexen-1-ol	402.1 ± 21.7 A	139.6 ± 9.2 C	346.0 ± 29.3 B	*	379.0 ± 8.2 A	293.4 ± 17.7 B	287.5 ± 25.4 B	*
cis 2-hexen-1-ol	258.5 ± 15.6 A	87.2 ± 8.8 B	225.8 ± 27.0 A	*	315.9 ± 11.6 A	243.8 ± 14.6 B	255.8 ± 10.4 B	*
*Alcohols*								
1-butanol	306.0 ± 44.1 A	196.0 ± 13.3 B	179.3 ± 7.4 B	*	262.4 ± 5.6 B	425.6 ± 36.6 A	174.6 ± 18.0 C	*
3-methyl-1-butanol	85.4 ± 7.7 B	59.5 ± 3.0 C	140.2 ± 11.5 A	*	35.8 ± 3.7 B	106.9 ± 15.3 A	126.5 ± 21.9 A	*
1-pentanol	15.8 ± 0.5 B	10.2 ± 0.4 B	18.3 ± 1.6 A	*	14.7 ± 1.0 C	26.6 ± 1.8 A	19.0 ± 1.2 B	*
1-octen-3-ol	9.6 ± 1.3 A	n.d. B	n.d. B	*	9.9 ± 0.6 A	5.8 ± 0.7 B	6.3 ± 1.3 B	*
1-heptanol	15.2 ± 0.2 A	13.3 ± 0.7 B	15.2 ± 0.5 A	*	6.0 ± 0.9 B	7.8 ± 0.4 A	5.7 ± 0.3 B	*
2-ethyl-1-hexanol	7.7 ± 0.9 B	2.9 ± 0.4 C	24.3 ± 3.5 A	*	n.d. C	3.5 ± 0.0 B	8.8 ± 1.2 A	*
1-octanol	n.d.	n.d.	n.d.		n.d. C	163.8 ± 10.0 B	252.3 ± 11.6 A	
*Terpenes*								
alpha-terpineol	3.7 ± 0.3 A	n.d. C	0.9 ± 0.0 B	*	3.3 ± 0.3 ns	3.1 ± 0.5 ns	4.4 ± 0.8 ns	ns
nerol	12.2 ± 0.4 B	n.d. C	16.7 ± 0.7 A	*	3.5 ± 0.9 AB	3.9 ± 0.6 A	2.7 ± 0.4 B	*
*Benzenoids*								
benzaldehyde	n.d.	n.d.	n.d.		n.d. B	6.6 ± 1.3 A	4.5 ± 1.1 A	*
benzyl alcohol	282.8 ± 13.7 B	248.8 ± 20.0 B	433.1 ± 50.5 A	*	20.0 ± 0.0 C	74.8 ± 9.3 A	44.1 ± 6.7 B	*
phenylethyl alcohol	82.9 ± 6.9 B	71.9 ± 3.7 B	126.6 ± 7.4 A	*	9.9 ± 0.4 C	27.8 ± 5.6 B	36.9 ± 7.0 A	*
syringol	n.d.	n.d.	n.d.		n.d. B	2.0 ± 0.3 A	n.d. B	
vanillin	89.3 ± 6.1 B	71.3 ± 2.6 B	108.4 ± 7.1 A	*	27.7 ± 3.4 A	13.3 ± 3.0 C	22.0 ± 2.5 B	*
methyl vanillate	n.d. B	11.1 ± 1.6 B	12.5 ± 2.1 A	*	n.d.	n.d.	n.d.	*
*Esters*								
butyl acetate	21.9 ± 1.2 A	n.d. C	1.6 ± 0.2 B	*	18.5 ± 0.6 A	2.7 ± 0.2 B	2.8 ± 0.5 B	*
*Acids*								
hexanoic acid	368.6 ± 20.7 A	223.8 ± 12.4 B	440.2 ± 66.5 A	*	443.8 ± 32.0 A	415.6 ± 63.1 A	280.3 ± 13.2 B	*

n.d. = not detected; * and different letters in the row of each compound refer to significant differences (Tukey; *p* < 0.05); ns = not significant.

**Table 2 foods-12-03661-t002:** ‘Nebbiolo’ bound VOCs detected in (a) skin and (b) pulp of ND (not defoliated/control), D1 (defoliated at fruit set—BBCH 71), and D2 (defoliated at berries touch—BBCH 81) samples. Mean concentration (expressed in µg/kg of berries) and standard deviation for each compound are reported.

	(a) SKIN				(b) PULP			
	ND	D1	D2	Sign.	ND	D1	D2	Sign.
*C6 compounds*								
trans 2-hexenal	71.8 ± 9.1 C	301.6 ± 10.1 A	227.0 ± 49.8 B	*	129.0 ± 8.8 B	105.7 ± 11.4 C	160.0 ± 9.7 A	*
1-hexanol	132.6 ± 5.6 C	380.9 ± 13.2 A	241.9 ± 46.8 B	*	74.1 ± 5.8 B	152.8 ± 21.3 A	199.7 ± 32.3 A	*
cis 3-hexen-1-ol	n.d. C	5.9 ± 0.6 A	4.5 ± 0.3 B	*	0.8 ± 0.1 B	1.4 ± 0.1 AB	2.0 ± 0.4 A	*
trans 3-hexen-1-ol	198.4 ± 17.3 B	362.8 ± 17.9 A	333.7 ± 83.0 A	*	151.8 ± 9.3 B	151.9 ± 13.5 B	269.0 ± 56.3 A	*
cis 2-hexen-1-ol	n.d.	n.d.	n.d.		1053.5 ± 132.2 B	1691.3 ± 225.9 B	2631.8 ± 721.9 A	*
*Alcohols*								
1-pentanol	21.2 ± 7.5 B	85.7 ± 6.9 A	25.0 ± 2.2 B	*	10.1 ± 2.0 B	21.1 ± 4.1 A	22.0 ± 3.7 A	*
1-butanol	462.8 ± 70.6 B	2944.6 ± 199.0 A	760.9 ± 244.7 B	*	541.4 ± 65.9 B	1470.0 ± 257.3 A	1045.2 ± 535.3 AB	*
3-methyl-1-butanol	132.9 ± 21.0 ns	227.1 ± 13.9 ns	225.3 ± 69.6 ns	ns	99.1 ± 15.9 B	99.7 ± 16.8 B	226.8 ± 84.7 A	*
1-heptanol	16.7 ± 0.2 B	29.4 ± 1.3 A	14.9 ± 2.3 B	*	3.7 ± 0.9 ns	4.8 ± 0.8 ns	4.5 ± 0.9 ns	ns
2-ethyl-1-hexanol	9.8 ± 2.1 B	14.8 ± 1.7 AB	26.5 ± 8.3 A	*	n.d. B	n.d. B	11.2 ± 3.1 A	*
1-octanol	16.4 ± 1.2 C	33.8 ± 1.3 A	21.5 ± 3.7 B	*	2.5 ± 0.2 B	3.2 ± 0.6 AB	5.1 ± 1.6 A	*
*Terpenes*								
cis linalooloxide	13.3 ± 1.7 B	23.9 ± 3.1 A	16.8 ± 2.3 B	*	3.3 ± 0.5 B	3.7 ± 0.5 AB	5.2 ± 1.0 A	*
trans linalooloxide	11.4 ± 2.0 B	22.7 ± 1.9 A	19.0 ± 3.9 A	*	3.7 ± 0.9 B	4.8 ± 0.8 AB	9.0 ± 3.4 A	*
nerol	9.5 ± 1.1 C	60.1 ± 2.0 A	24.7 ± 6.6 B	*	n.d.	n.d.	n.d.	
*Benzenoids*								
benzaldehyde	8.2 ± 2.9 B	20.0 ± 2.3 A	13.6 ± 3.3 B	*	4.2 ± 1.0 B	3.3 ± 0.4 B	5.8 ± 1.2 A	*
benzyl alcohol	1990.3 ± 139.3 C	6340.9 ± 154.5 A	3800.2 ± 1238.7 B	*	2027.2 ± 187.5 B	2867.3 ± 165.0 AB	4283.9 ± 1327.0 A	*
phenylethyl alcohol	304.1 ± 22.3 B	631.2 ± 10.6 A	533.4 ± 169.7 A	*	217.3 ± 18.9 B	247.4 ± 14.2 B	414.5 ± 102.0 A	*
syringol	27.3 ± 3.2 B	43.7 ± 5.3 A	36.7 ± 8.4 A	*	n.d.	n.d.	n.d.	
vanillin	77.2 ± 5.9 B	139.0 ± 7.4 A	55.4 ± 19.5 B	*	8.5 ± 2.4 B	31.7 ± 2.7 A	39.0 ± 10.1 A	*
methyl vanillate	90.3 ± 7.3 B	134.6 ± 4.7 AB	141.2 ± 34.8 A	*	15.5 ± 2.0 B	23.5 ± 1.8 B	45.0 ± 14.4 A	*

n.d. = not detected; * and different letters in the row of each compound refer to significant differences (Tukey; *p* < 0.05); ns = not significant.

**Table 3 foods-12-03661-t003:** ‘Aleatico’ free VOCs detected in (a) skin and (b) pulp of ND (not defoliated/control), and D1 (defoliated at fruit set—BBCH 71) samples. Mean concentration (expressed in µg/kg of berries) and standard deviation for each compound are reported.

	(a) SKIN			(b) PULP		
	ND	D1	Sign.	ND	D1	Sign.
*C6 compounds*						
hexanal	7764.2 ± 4594.9 ns	6199.6 ± 284.5 ns	ns	623.6 ± 244.9 ns	422.9 ± 4.7 ns	ns
trans 2-hexenal	1987.6 ± 209.4 ns	2804.5 ± 58.2 ns	ns	425.3 ± 222.9 ns	375.2 ± 22.7 ns	ns
2-hexanol	80.1 ± 14.6 ns	82.0 ± 3.5 ns	ns	46.9 ± 5.0 ns	47.5 ± 2.9 ns	ns
1-hexanol	342.9 ± 4.9 B	597.2 ± 4.7 A	*	179.2 ± 17.6 A	6.9 ± 1.6 B	*
trans-3-hexen-1-ol	245.3 ± 0.4 B	831.7 ± 21.2 A	*	180.0 ± 23.6 ns	199.8 ± 10.9 ns	ns
cis-2-hexen-1-ol	261.0 ± 2.9 B	486.7 ± 22.2 A	*	29.3 ± 18.6 ns	40.0 ± 3.4 ns	ns
*Alcohols*						
1-butanol	78.6 ± 1.6 ns	204.0 ± 69.6 ns	ns	50.2 ± 7.7 ns	165.1 ± 158.0 ns	ns
3-methyl-1-butanol	227.6 ± 0.4 A	190.0 ± 0.9 B	*	133.8 ± 8.9 A	61.5 ± 2.0 B	*
1-pentanol	72.5 ± 0.4 ns	71.0 ± 4.7 ns	ns	26.2 ± 1.1 A	16.3 ± 0.6 B	*
1-heptanol	131.7 ± 2.2 ns	152.4 ± 4.5 ns	ns	36.6 ± 6.7 ns	30.1 ± 0.5 ns	ns
1-octanol	1646.5 ± 32.8 ns	2686.7 ± 251.2 ns	ns	57.3 ± 1.5 ns	52.4 ± 0.6 ns	ns
2-ethyl-1-hexanol	150.2 ± 11.3 ns	147.6 ± 6.7 ns	ns	146.5 ± 2.0 ns	135.4 ± 14.5 ns	ns
1-octen-3-ol	42.0 ± 3.2 ns	38.9 ± 5.2 ns	ns	56.8 ± 12.6 ns	41.4 ± 1.0 ns	ns
*Terpenes*						
beta-myrcene	83.1 ± 0.1 ns	68.6 ± 6.3 ns	ns	20.0 ± 3.7 A	12.4 ± 0.3 B	*
limonene	68.7 ± 7.6 ns	66.6 ± 32.6 ns	ns	n.d.	n.d.	
cis-linalool oxide	n.d.	n.d.		65.2 ± 15.4 ns	38.7 ± 1.8 ns	ns
trans-linalool oxide	n.d.	n.d.		42.8 ± 4.6 ns	60.8 ± 14.0 ns	ns
beta-linalool	609.2 ± 62.1 ns	824.6 ± 74.0 ns	ns	125.6 ± 17.1 A	113.2 ± 12.9 B	*
alpha-terpineol	72.0 ± 5.9 B	100.9 ± 7.5 A	*	29.4 ± 4.9 ns	15.1 ± 6.1 ns	ns
epoxylinalool	78.8 ± 5.7 ns	81.1 ± 12.9 ns	ns	40.5 ± 10.6 ns	34.0 ± 2.0 ns	ns
nerol	497.0 ± 13.4 ns	535.3 ± 2.8 ns	ns	38.3 ± 4.2 ns	24.5 ± 0.6 ns	ns
geranic acid	6358.8 ± 398.2 ns	6115.8 ± 439.2 ns	ns	306.0 ± 14.7 A	125.6 ± 6.7 B	*
*Benzenoids*						
benzaldehyde	49.0 ± 2.3 ns	51.8 ± 5.5 ns	ns	23.4 ± 0.5 ns	20.8 ± 0.9 ns	ns
benzyl alcohol	255.0 ± 15.6 ns	227.7 ± 1.8 ns	ns	150.6 ± 5.9 A	80.5 ± 2.8 B	*
phenylethyl alcohol	348.7 ± 2.1 ns	391.0 ± 30.0 ns	ns	161.1 ± 12.6 A	105.1 ± 0.2 B	*
vanillin	30.8 ± 55.9 ns	160.6 ± 4.0 ns	ns	228.8 ± 58.1 ns	184.5 ± 43.1 ns	ns
*Esters*						
methyl butyrate	1298.5 ± 26.1 B	1595.2 ± 31.3 A	*	590.9 ± 59.5 A	450.8 ± 5.9 B	*
butyl acetate	30.8 ± 43.6 ns	167.5 ± 10.4 ns	ns	31.7 ± 20.0 ns	38.3 ± 1.1 ns	ns
ethyl decanoate	350.7 ± 20.3 A	191.9 ± 5.0 B	*	135.3 ± 1.2 A	103.7 ± 4.2 B	*
*Acids*						
hexanoic acid	4007.4 ± 1313.1 ns	4550.5 ± 379.7 ns	ns	829.9 ± 596.3 ns	558.5 ± 0.6 ns	ns
2-hexenoic acid	n.d.	n.d.		77.4 ± 29.8 ns	64.6 ± 6.5 ns	ns
octanoic acid	n.d.	n.d.		100.9 ± 107.5 ns	30.2 ± 10.0 ns	ns
nonanoic acid	90.8 ± 16.8 ns	92.0 ± 6.0 ns	ns	173.1 ± 157.2 ns	51.6 ± 5.0 ns	ns
dodecanoic acid	253.8 ± 56.4 ns	259.4 ± 58.1 ns	ns	23.3 ± 9.8 ns	21.1 ± 6.1 ns	ns
*Other compounds*						
furfural	413.8 ± 12.5 ns	284.5 ± 29.2 ns	ns	600.8 ± 266.7 A	250.7 ± 31.9 B	*

n.d. = not detected; * and different letters in the row of each compound refer to significant differences (Tukey; *p* < 0.05); ns = not significant.

**Table 4 foods-12-03661-t004:** ‘Aleatico’ bound VOCs detected in (a) skin and (b) pulp of ND (not defoliated/control), and D1 (defoliated at fruit set—BBCH 71) samples. Mean concentration (expressed in µg/kg of berries) and standard deviation for each compound are reported.

	(a) SKIN			(b) PULP		
	ND	D1	Sign.	ND	D1	Sign.
*C6 compounds*						
hexanal	45.9 ± 6.6 B	94.2 ± 0.1 A	*	28.4 ± 1.0 ns	21.4 ± 0.1 ns	ns
trans 2-hexenal	222.5 ± 34.3 B	418.1 ± 22.5 A	*	86.0 ± 1.0 ns	118.7 ± 0.3 ns	ns
cis 3-hexen-1-ol	28.8 ± 40.7 ns	28.0 ± 9.0 ns	ns	14.0 ± 0.4 A	10.2 ± 1.6 B	*
2-hexanol	253.1 ± 79.1 ns	362.8 ± 15.5 ns	ns	133.9 ± 1.3 A	107.9 ± 3.7 B	*
1-hexanol	264.6 ± 88.1 ns	408.4 ± 8.7 ns	ns	233.7 ± 2.7 A	144.6 ± 5.3 B	*
trans-3-hexen-1-ol	194.8 ± 9.3 B	350.2 ± 1.1 A	*	200.7 ± 0.9 A	177.9 ± 3.7 B	*
cis-2-hexen-1-ol	68.5 ± 65.8 ns	58.0 ± 57.3 ns	ns	20.5 ± 0.1 A	9.2 ± 0.9 B	*
*Alcohols*						
isobutyl alcohol	n.d.	n.d.		24.2 ± 0.1 A	13.0 ± 1.0 B	*
3-pentanol	28.9 ± 6.8 ns	47.0 ± 17.1 ns	ns	14.5 ± 2.0 ns	7.6 ± 3.2 ns	ns
2-pentanol	47.7 ± 67.5 ns	41.4 ± 9.2 ns	ns	6.7 ± 1.0 ns	8.5 ± 2.0 ns	ns
1-butanol	106.9 ± 89.3 ns	149.2 ± 23.9 ns	ns	93.1 ± 1.3 A	58.3 ± 3.3 B	*
3-methyl-1-butanol	105.2 ± 32.1 ns	137.2 ± 27.4 ns	ns	180.1 ± 3.3 A	116.5 ± 0.2 B	*
3-methyl-2-buten-1-ol	353.5 ± 70.9 B	624.1 ± 11.4 A	*	468.7 ± 0.4 A	368.4 ± 16.0 B	*
1-heptanol	84.0 ± 3.7 ns	82.7 ± 5.0 ns	ns	38.3 ± 0.5 A	28.0 ± 0.3 B	*
1-octanol	1712.4 ± 2398.0 ns	23.3 ± 3.5 ns	ns	n.d.	n.d.	
3-octanol	59.0 ± 51.5 ns	25.3 ± 1.8 ns	ns	9.8 ± 0.6 ns	6.6 ± 1.0 ns	ns
2-ethyl-1-hexanol	64.2 ± 27.3 ns	40.7 ± 4.1 ns	ns	15.4 ± 0.2 ns	13.3 ± 2.0 ns	ns
1-octen-3-ol	0.0 ± 0.0 ns	21.9 ± 31.0 ns	ns	13.2 ± 0.2 ns	10.9 ± 0.1 ns	ns
*Terpenes*						
beta-myrcene	5598.4 ± 3169.5 ns	2444.4 ± 103.9 ns	ns	627.2 ± 4.9 A	322.0 ± 33.4 B	*
limonene	1982.1 ± 1351.0 ns	439.0 ± 9.0 ns	ns	197.4 ± 18.4 ns	75.5 ± 34.7 ns	ns
beta-trans-ocimene	511.6 ± 376.2 ns	483.6 ± 25.7 ns	ns	141.2 ± 5.7 A	65.7 ± 11.4 B	*
beta-cis-ocimene	1779.3 ± 522.4 ns	1006.2 ± 1307.1 ns	ns	668.7 ± 11.1 A	542.6 ± 43.4 B	*
cis-linalool oxide	498.0 ± 83.9 ns	552.7 ± 218.8 ns	ns	6.2 ± 0.2 B	183.0 ± 0.3 A	*
trans-linalool oxide	43.5 ± 5.5 B	59.9 ± 4.9 A	*	26.4 ± 0.1 A	19.6 ± 0.5 B	*
beta-linalool	2875.5 ± 1244.0 ns	5033.0 ± 321.6 ns	ns	1520.5 ± 2.6 A	958.5 ± 21.2 B	*
cis-citral	210.9 ± 68.4 B	394.1 ± 44.1 A	*	91.1 ± 0.0 A	60.6 ± 5.6 B	*
alpha-terpineol	186.2 ± 37.1 ns	307.7 ± 10.7 ns	ns	82.7 ± 0.3 A	58.1 ± 0.5 B	*
trans-citral	288.3 ± 149.2 B	553.3 ± 69.1 A	*	n.d.	n.d.	
epoxylinalool	361.5 ± 124.6 ns	530.9 ± 0.6 ns	ns	151.0 ± 1.1 A	119.0 ± 1.1 B	*
citronellol	106.4 ± 0.5 B	147.8 ± 10.8 A	*	n.d.	n.d.	
cis-geraniol	2457.8 ± 1048.5 B	4996.2 ± 109.8 A	*	1068.1 ± 18.5 A	681.2 ± 58.2 B	*
trans-geraniol	6653.7 ± 3249.0 B	13920.3 ± 211.6 A	*	2577.5 ± 48.4 A	1493.2 ± 127.6 B	*
geranic acid	27168.5 ± 5493.3 ns	34453.0 ± 222.4 ns	ns	5928.6 ± 7.7 A	3461.4 ± 317.6 B	*
8-hydroxylinalool	66.7 ± 19.6 ns	45.2 ± 24.9 ns	ns	13.4 ± 0.9 ns	29.7 ± 29.7 ns	ns
*Benzenoids*						
eugenol	145.6 ± 38.9 ns	114.2 ± 22.8 ns	ns	57.2 ± 0.3 A	38.3 ± 7.0 B	*
2-methoxy-4-vinylphenol	75.2 ± 73.5 ns	22.0 ± 3.9 ns	ns	n.d.	n.d.	
methyl vanillate	449.4 ± 228.6 ns	422.6 ± 0.2 ns	ns	83.9 ± 1.4 ns	75.2 ± 17.6 ns	ns
acetovanillone	523.1 ± 24.7 B	661.4 ± 46.4 A	*	195.4 ± 2.4 ns	49.3 ± 42.8 ns	ns
4-hydroxy-3-methylacetophenone	75.2 ± 44.3 ns	89.1 ± 13.8 ns	ns	n.d.	n.d.	
benzaldehyde	73.7 ± 15.0 ns	54.2 ± 4.0 ns	ns	34.5 ± 0.2 ns	38.1 ± 13.9 ns	ns
benzyl alcohol	1362.7 ± 605.8 ns	2108.8 ± 222.0 ns	ns	1664.4 ± 6.4 A	1158.0 ± 102.1 B	*
phenylethyl alcohol	1878.8 ± 844.1 ns	3150.8 ± 261.0 ns	ns	1570.0 ± 4.8 A	1059.4 ± 88.8 B	*
*Esters*						
methyl butyrate	229.5 ± 117.3 ns	196.2 ± 6.1 ns	ns	60.8 ± 0.2 ns	52.0 ± 5.1 ns	ns
butyl acetate	221.1 ± 7.1 A	58.0 ± 12.2 B	*	59.6 ± 3.3 ns	63.8 ± 1.4 ns	ns
*Acids*						
hexanoic acid	90.3 ± 67.0 ns	64.3 ± 39.4 ns	ns	24.4 ± 0.6 A	10.8 ± 0.2 B	*
*Other compounds*						
3-hydroxy-beta-damascone	591.1 ± 156.6 ns	778.4 ± 56.5 ns	ns	328.8 ± 0.9 ns	913.5 ± 778.2 ns	ns

n.d. = not detected; * and different letters in the row of each compound refer to significant differences (Tukey; *p* < 0.05); ns = not significant.

## Data Availability

Data is contained within the article.

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
