# Peer review of "Effects of Leaf Removal on Free and Glycoconjugate Aromas of Skins and Pulps of Two Italian Red Grapevine Varieties"

_foods, 2023, doi:10.3390/foods12193661_

Round 1

Reviewer 1 Report

In this manuscript authors investigated the impact of leaf removal at two phenological stages on free and glycoconjugate aromas of skins and pulps of two grapevine varieties. The paper is moderately well written and the thematic is still actual, but it needs substantial improvement if intended to be published in Foods journal.

One of the major flaws of the manuscript is the fact that the research was conducted during only one season, and this is not enough to make solid conclusion in viticulture research. In this regard authors are encouraged to include at least one more year of data if they have them. The English language and the style of writing should be improved. Specific comments about the manuscript are given bellow:

Lines 2-4: The title should be improved. Delete the word ‘bunches’. Use the term ‘grapevine varieties’ instead of ‘grapes’. Leaf removal was not done at two phenological stages on both varieties, so this should be adapted.

Line 15: use the term ‘fruit zone microclimate’ (or ‘cluster zone microclimate’) instead of ‘cluster microclimate’ (here and on other positions in the text).

Line 16: The term ‘therefore’ is not appropriate, as it is not implied that leaf removal has this effect (in some cases it has the opposite effect).

Line 23: Do not use the term ‘more resilient’ in this context.

Line 24: Use the term ‘grapevine varieties’ instead of ‘grapes’ (here and on other positions in the text).

Line 37: the same as in line 16, the term ‘therefore’ is not appropriate, as it is not implied that leaf removal has this effect (in some cases it has the opposite effect).

Lines 50-52: If you have the references supporting these two assumptions then please insert them in the text. Otherwise delete this text or make it clear that it is just a hypothesis by the authors.

Lines 73-80: Delete this text, as leaf removal for this purpose is done in the apical part of the canopy, and not in the cluster zone.

Introduction and Discussion sections: Additional papers published in the last five years on the thematic of the impact of leaf removal on aromas in grapes and wines should be added.

Lines 88-90: At least some basic information about the type of defoliation which was performed should be indicated here, in order to avoid the need to consult another paper to get this information. Moreover, it is not indicated the year when this was performed.

Line 90: Do not use the term ‘and corroborates’ and consult the Guide to Authors in this regard.

Lines 92-94: Indicate the dates when leaf removal was performed.

Lines 95-101: This part is very unclear and incomplete, especially regarding the blocks and the three parts of the vineyard, so it should be carefully rewritten. Moreover, it is not indicated the data for Aleatico variety.

Line 104: Harvest dates should be indicated.

Lines 164-167: Delete this part as this was already stated.

Lines 167-178: Move this part to the Introduction section and make it shorter.

Line 179: Use ‘concentration’ instead of ‘total amount’.

Line 180: Use ‘total amount’ instead of ‘’global quantity’.

Line 182: Change ‘off’ to ‘of’.

Line 184: Add ‘were present’.

Lines 185-187: Delete this sentence as this was already stated.

Line 188: It is not indicated the direction of the mentioned significant differences (increase, decrease,…).

Line 192: It is not indicated the direction of the mentioned greatest impact (increase, decrease,…).

Lines 193-194 (and on other positions in the text): Do not use the term ‘stress’ in this context unless you have measured that some stress occurred. The removal of some leaves does not necessarily imply that some stress occurred on the plant.

Table 1: Use ‘Sign.’ or ‘Significance’ instead of ‘ND vs D1 vs D2’ (here and in other tables).

Lines 317 and 332: Do not use the term ‘looking’.

Lines  358-384: Delete this text as it is just an assumption, which is not based on the results. The extraction and modification of volatile compounds and their precursors during maceration are not so linear and simple to make solid assumptions and conclusions in this regard.

The text should be revised for English language and some indications in this regard are given in the Specific comments.

Reviewer 2 Report

In the present manuscript, the effects of leaf removal at different stages on the free and bound aromas of skins and pulps of two Italian red grapes were studied. This research were organized well and expanded our knowledge in viticulture. However, there were still some problems or errors in this manuscript. Thus, we gave a minor review to this work. Some comments or suggestions were as following:

1, As the grape cultivar, Nebbiolo should be enclosed in single quotation marks, as Nebbiolo. Other grape cultivar names should also be arranged in this way.

2, How many vintages did the authors choose to have their field experiment? In recent years, the climates all over the word changed greatly year by year. Thus, only one years study limited the information it could offered to the readers.

3, The introduction of the defoliation was in detail. However, how many biological duplication did the authors collected for each treatment or sample?

4, Line 114, 10000 g should be revised as 10,000 × g. And more, the letter g should be in italic.

5, Line 141, m/z should be in italic. Line 160, letter p in p<0.05 should also be in italic.

6, In Figure 1, we could not see standard deviation in these data. Similar problems also occurred in Figure2.

7, The format of some references were not standard, for example, 11, 34 and etc.

Reviewer 3 Report

This research focused on the leaf removal at different stage with aim to improve the aroma of grapes. The results is interesting. There are several issues that need to be addressed.

1)      Title: should the 'bounches' be 'bunches'?

2)      Materials and Methods: defoliation was performed at two phenological stages for Nebbiolo grapes, whereas only one treatment was applied for Aleatico grapes. Why?

3)      Line 103-104: the berries of all treatments reached a TSS of 23±1°Brix at the same time (BBCH 89)?

4)      By comparing Figure 1 and Figure 2, it can be observed that the two varieties showed different response to the similar treatment (leaf removal at fruit set). This phenomenon and the underlying mechanism should be analyzed and discussed.

5)      Line 393-396: only one treatment at fruit set stage (D1) was conducted for Aleatico grapes, while the treatment at conducted at berries touch (D2) for Nebbiolo grapes showed a significant effect (line 393-395). These two results are not comparable, and it may be not sure that leaf removal at other stage also shows less evident impact on VOCs accumulation for Aleatico grapes.

Line 16: such as volatiles.

Line 66: change 'as such as' to 'such as'?

Round 2

Reviewer 1 Report

All the issues have been adressed and I suggest to accept the manucript.